# GRAPH SCHEMAS AS ABSTRACTIONS FOR TRANSFER LEARNING, INFERENCE, AND PLANNING

## ABSTRACT

We propose schemas as a model for abstractions that can be used for rapid transfer learning, inference, and planning. Common structured representations of concepts and behaviors—schemas—have been proposed as a powerful way to encode abstractions. Latent graph learning is emerging as a new computational model of the hippocampus to explain map learning and transitive inference. We build on this work to show that learned latent graphs in these models have a slot structure—schemas—that allow for quick knowledge transfer across environments. In a new environment, an agent can rapidly learn new bindings between the sensory stream to multiple latent schemas and select the best fitting one to guide behavior. To evaluate these graph schemas, we use two previously published challenging tasks: the memory & planning game and one-shot StreetLearn, that are designed to test rapid task solving in novel environments. Graph schemas can be learned in far fewer episodes than previous baselines, and can model and plan in a few steps in novel variations of these tasks. We further demonstrate learning, matching, and reusing graph schemas in navigation tasks in more challenging environments with aliased observations and size variations, and show how different schemas can be composed to model larger environments.

## 1 INTRODUCTION

Discovering and using the right abstractions in new situations affords efficient transfer learning as well as quick inference and planning. Common reusable structured representations of concepts or behaviors—schemas—have been proposed as a powerful way to encode abstractions (Tenenbaum et al., 2011; Mitchell, 2021)(Fig.1A). Humans excel at this ability, argued to be a key factor behind intelligence and a fundamental limitation in current AI systems (Shanahan & Mitchell, 2022). A computational model with the ability to discover and reuse previously-learned structured knowledge to behave and plan in novel situations can advance AI systems.

Experimental evidence suggests that some animals have this ability. Rats and mice tend to learn new environments faster if they can reuse past schemas (Zhou et al., 2021; Tse et al., 2007), and there are cells in the macaque hippocampus encoding schemas for spatial abstractions (Baraduc et al., 2019). Neural circuits in hippocampus (HPC) and prefrontal cortex (PFC) are both implicated in different aspects of schema learning, recognition, update, and maintenance. Recent work in mice suggested that PFC encodes the schema while HPC binds this schema to sensorimotor specifics of a task (Samborska et al., 2022). New experiences are also shown to be learned within a single trial if it fits an existing schema, and such knowledge rapidly becomes HPC-independent much faster than typically expected for memory consolidation. Kumaran et al. (2016) proposed an updated complementary learning systems (CLS) theory based on this evidence, but there is so far no explicit demonstration of such rapid learning with schema reuse as far as we know. Schema-based mechanisms also aid in memory consolidation over longer time-scales into reusable knowledge structures (Gilboa & Marlatte, 2017). Even though spatial navigation and memory dominated research in HPC,it is also known to be important for non-spatial declarative memory and relational reasoning tasks. Structured relational representations have been proposed as a common mechanism that generalizes to spatial and non-spatial tasks and memory (Eichenbaum & Cohen, 2014; Stachenfeld et al., 2017). Recent work on cognitive maps in HPC model these representations using higher order latent graph structures and show generalization to disparate HPC functions (Whittington et al., 2020; George et al., 2021; Sharma et al., 2021; Whittington et al., 2021).

In this work, we take one such model and provide a concrete computational model of abstractions using graph schemas. We describe how graph schemas can be learned and then reused for transfer learning, quick inference, and planning for behavior in new situations by rapidly learning observation bindings and discovering the best schema online. We build these schemas using clone-structured cognitive graphs (CSCG), a computation model of cognitive maps in HPC (George et al., 2021). In particular, CSCG work showed that latent high order graphs can be learned in highly aliased settings using a smooth, probabilistic, parameterization of the graph learning problem using gradient-based optimization. We use navigation as our setting, which requires handling perceptual aliasing, a difficult problem in which standard methods fail (Lajoie et al., 2018). Using higher order graphs helps with modeling aliased observations (Xu et al., 2016). CSCGs also generalize beyond spatial navigation to non-spatial domains (Dedieu et al., 2019), so schemas on CSCG can extend beyond spatial navigation. This generalization of graph schemas neatly maps to the idea of useful abstractions as template learning and structure mapping as described in Shanahan & Mitchell (2022). Our computational model provides a concrete implementation of some of the operations described in that work. For example, their operation of seeing similarity can be interpreted as the process of finding which of the known templates best describes the observations in terms of likelihood in our model, and binding different instantiations to nodes in a structured graph can be interpreted as learning different emission bindings in a graph schema.

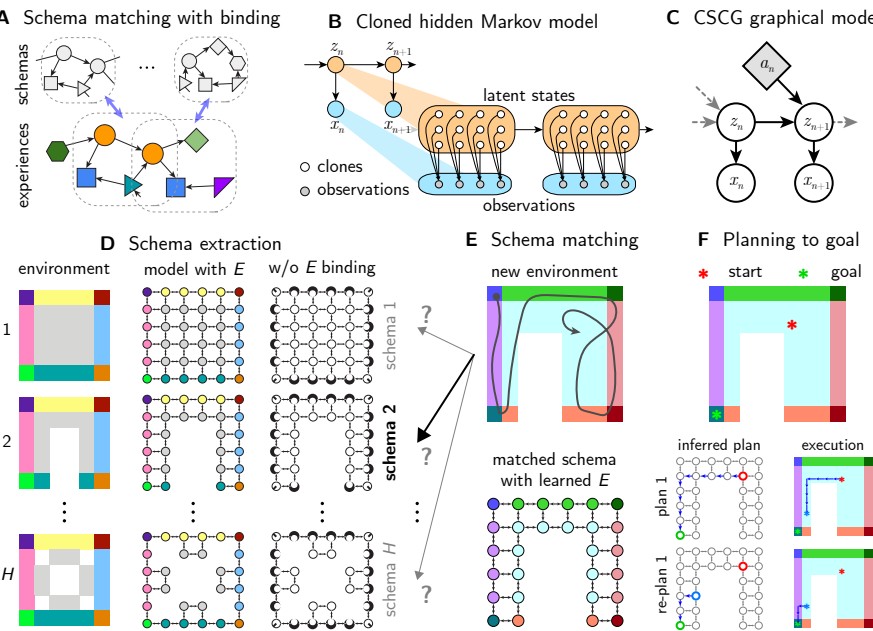

Figure 1: Overview of using graph schemas in CSCG model. **A.** Schemas serve as abstractions that can be matched with new experiences. Multiple schemas can be composed together to model observations that could be aliased. **B.** Schematic of cloned HMM with multiple cloned states sharing the emissions to model them in different contexts. **C.** A graphical model of CSCG with action conditioning. **D.** Schemas in CSCG are extracted from learned models with transitions and emissions, and then unbinding the specific emissions (shown as colors) but keeping the clone structure (shown as node shapes). **E.** In a novel environment with different observations, the agent navigates and finds the schema that best explains the observations by learning new bindings. **F.** Planning can be performed in a new room using the matched schema. As the plan is being executed, new observations are used to update the beliefs about current location and re-plan as needed.

## 2 RELATED WORK

Large language models (LLMs) have shown impressive demonstrations of few-shot generalization to new tasks without re-training, possibly using in-context learning (Brown et al., 2020; Wei et al.,

2021). While LLMs ostensibly use some form of abstractions and rapid binding to new inputs, the exact mechanism for this is an active research area (Min et al., 2022).

In a navigation setting, rapid generalization for solving tasks in novel environments can be broken down into three stages: exploration, modeling, and planning. Different approaches treat these stages as either separate or entangle them in various combinations. Some of the recent work on learning to generalize to novel tasks and environments is predominantly done in a reinforcement learning (RL) framework. Approaches using model-free RL show some generalization and behavior that mimics model-based RL agents in some settings, but are limited in their generalization (Wang et al., 2016). More recent works such as episodic planning network (EPN) and hierarchical chunk attention memory (HCAM) added episodic memory storage with attention heads to selectively attend and reuse stored memories to rapidly adapt to solving tasks in new environments in few-shot settings (Ritter et al., 2020; Lampinen et al., 2021). These models match optimal planning only after training for billions of steps. Further, none of these approaches explicitly builds models of the environment and plans over them. Gupta et al. (2017) show that using explicit model building (mapping) and navigation via planning in spatial environments can handle partial observability and outperform methods without this ability. In most, if not all, of these settings, observations are not aliased and therefore unique. Model-based RL systems do build and plan on explicit models, however they typically start with a known model or require extensive training data to build the model (Moerland et al., 2020). By adding more structure to the model and the ability to reuse the learned structures in novel environments, training can potentially be done in fewer samples.

Schema matching is another related area where some recent work focuses on finding correspondence between graphs in different contexts. Work by Crouse et al. (2021) was the first to use the neural network approach to structure mapping, but is restricted to the matching problem and has no active exploration to resolve or learn new structures or to plan with partially matching schemas. Another line of related work focused on solving simplified relational tasks inspired by Raven's Progressive Matrices (Webb et al., 2021; Kerg et al., 2022). The main idea is to separate abstract relations from sensory observations during training and learn the observation mapping to solve new tasks with the same relations but novel mapping, but in a deterministic and simplified setting. As far as we know, ours is the first model to learn explicit latent higher-order graph structure to model the environment only from observation sequences, and to build schemas on top of that model. These allow rapid transfer, inference, and planning in problems that can be cast as (stochastic) graph navigation.

## 3 Preliminaries

Consider an agent navigating in a directed graph. When the agent visits a node in that graph, the node emits an observation. However, multiple nodes may emit the same observation (i.e., they are *aliased*), so this is not enough to disambiguate where in the graph the agent is located. Additionally, actions do not have deterministic results—executing the same action at the same node might result in the agent navigating to different nodes. The outgoing edges from a node are labeled with the action that would traverse them, and with the probability of traversing them under that action. The sum of the probabilities of all outgoing edges from a node with the same action label sum up to 1. We call this graph $G$, which defines the environment.

When the agent performs a sequence of actions $a_1, \ldots, a_N$ (with discrete $a_n \in \{1, \ldots N_{\text{actions}}\}$), it will as a result receive a sequence of observations, obtaining the stream $x_1, \ldots, x_N$ (with discrete $x_n \in \{1, \ldots N_{\text{obs}}\}$). The goal of learning is to recover the topology of the environment $G$ from sequences of actions and observations. The goal of transfer is to reuse the topology of the graph when the nodes are relabeled with new observations with or without structure-preservation. Additionally, we can exploit the learned graphs for further tasks, such as goal-directed navigation.

### 3.1 Clone-Structured Cognitive Graphs

Clone-structured cognitive graphs (CSCGs) were introduced in Dedieu et al. (2019); George et al. (2021) to recover (an approximation of) the graph $G$ from sequences of action-observation pairs. To do this, they use a categorical hidden variable $z_1, \ldots, z_N$ to model the node of the graph that the agent is at in each time step. With this, it is possible to formulate a graphical model for the sequence of observations given the actions. If we take the conditional version of their model, but write the emission probabilities explicitly, the connection with hidden Markov models (HMMs)

becomes more obvious:

$$P(x|a) = \sum_z P(z_1) \prod_{n=2}^{N} P(z_n|z_{n-1}, a_{n-1}) \prod_{n=1}^{N} P(x_n|z_n), \tag{1}$$

where we use the shorthand $x \equiv \{x_1 \ldots, x_N\}$, $a \equiv \{a_1 \ldots, a_N\}$, $z \equiv \{z_1 \ldots, z_N\}$. Additionally, we define a *transition matrix* $T$ with elements $T_{ijk} = P(z_n = k|z_{n-1} = j, a_{n-1} = i)$; and an *emission matrix* $E$ with elements $E_{ij} = P(x_n = j|z_n = i)$.

For an unrestricted $E$, this is an action-conditional version of an HMM. What makes this a CSCG is the structure of the emission matrix: each row contains a single 1 and all the remaining entries are zero—equivalently, emissions are deterministic. Multiple hidden states can emit the same observation, however. All the hidden states that emit the same observation are called the *clones* of that observation. Also, unlike HMMs, learning proceeds by choosing a particular allocation of clones to observations, encoding it in $E$, and keeping $E$ fixed throughout learning.

Learning a new graph $G$ describing a particular environment only involves learning $T$, i.e. maximizing $P(x|a; T, E)$ wrt $T$ (See Appendix A.1 for details). Transfer learning between different environments is possible too: if we know that a new environment has the same topology as a previous one for which $T$ has already been learned, but with different emissions associated to its nodes, we can maximize $P(x|a; T, E)$ wrt $E$ while keeping $T$ fixed to its known value from the previous environment (See Appendix A.2 for details). Further, if we know that the new environment preserves the emission structure (when two nodes emitted the same observation in the original environment, they also do so in the new environment), then we can further restrict the learning of $E$, with all the rows of $E$ that correspond to the same observation in the original environment sharing the same parameters. Learning can be efficiently accomplished using the EM algorithm, analogous to learning in HMMs, but with some computational advantages derived from the structure of $E$. The graphical model is depicted in Fig. 1B-C.

## 3.2 SCHEMAS FOR TRANSFER LEARNING

As explained in the previous subsection, it is possible to reuse parts of a previously learned CSCG to learn a new environment faster. The portion of the CSCG that gets transferred across environments is called a *schema*, which consists of the transition matrix together with its unbinded emission structure (Fig. 1D). For example, in the room navigation setting, a schema models the locations in a room and how actions move the agent between them, as well as the knowledge that the floor or door can look the same at multiple locations in the room. Schemas allow for rapid model learning in new environments that have matching topologies and emission structures with fast binding (Fig. 1E). Inference can be performed with the matched schemas to actively plan and pursue goals. We can also detect transitions to another known schema, or to unknown territory by comparing likelihoods of observations under different schemas.

## 4 RESULTS

### 4.1 MEMORY PLANNING GAME

We first evaluate our model on the Memory & Planning Game (MPG) proposed by Ritter et al. (2020) to evaluate rapid adaptation to new environments and solve tasks in that environment. In the game, the agent can navigate on a $4 \times 4$ grid, observing symbols, and the task is to collect rewards at a specified goal location in that grid (Fig. 2A). After collecting a reward, the agent is re-spawned in a new location and a new goal is specified. All grid positions have unique symbols and the symbol-position mapping is randomized after each episode, which lasts for 100 steps. See B.1 for details. This setup lets us evaluate our model and schema reuse as the structure is maintained across episodes and the agent needs to explore to collect the observations and bind them rapidly to the schema to maximize rewards with optimal navigation.

**CSCG schema learns the graph structure in few episodes.** The CSCG schema agent first explores the grid randomly collecting observations for a few episodes. After each episode, we learn a CSCG model that best explains the experiences across all episodes so far observed. We reuse the same schema ($T$) across all episodes and learn a new binding (emission matrix $E$) per episode. It takes

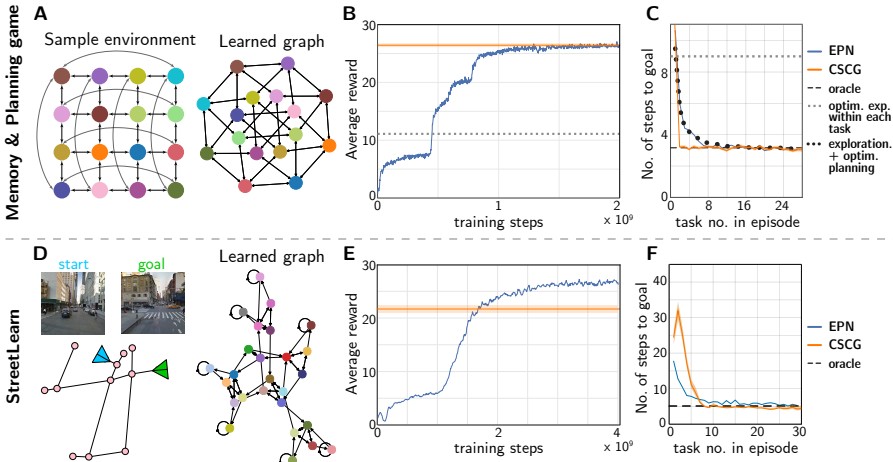

Figure 2: Performance on MPG and One-shot StreetLearn. **A.** An example MPG episode and the latent graph learned by the CSCG. **B.** Average reward in MPG. CSCG performance remains the same after 9 episodes (900 steps), so we plotted the average. EPN performance is shown on evaluation at different stages of training. **C.** Number of steps to get the reward in subsequent tasks in an episode. CSCG explores first without collecting the reward, so our first task takes longer to complete but subsequent tasks do optimal planning. **D.** Example One-Shot StreetLearn environment, with agent's observations. Right panel shows the latent graph learned by CSCG. **E.** Average reward across 100 episodes of StreetLearn in the test city. Notation is similar to MPG plot. **F.** Number of steps to get the reward in subsequent tasks in an episode. First few tasks take variable numbers of steps as we collect rewards incidentally. After that, the CSCG agent collects all subsequent rewards in optimal steps. EPN baselines and optimal values for both environments are re-plotted using the data from Ritter et al. (2020). Error bars are 95% CI of the SEM.

only 9 episodes (900 steps) to learn a perfect schema of this $4 \times 4$ grid environment and in subsequent episodes, we rapidly learn the schema bindings and do planning to maximize the rewards. Learning an efficient exploration policy to cover all observations in a new episode is an interesting problem, but, since our focus in this work is schema learning and reuse, we employ two different hard coded exploration policies: random navigation actions and an optimal set of actions to cover the learned transition graph. Average reward per episode $\pm$ standard error of the mean (SEM) after learning the schema is: $17.3 \pm 0.57$ for random and $26.4 \pm 0.17$ for optimal exploration policy, which is comparable to Episodic planning network (EPN) ($\sim$26)(Ritter et al., 2020). In contrast, EPN takes more than 10 million episodes ($> 10^9$ steps) of training to reach its optimal performance (Fig. 2B). Planning in our model is optimal, on par with EPN and the oracle. Fig. 2C shows the average number of steps taken to collect rewards in subsequent tasks. CSCG performance remains the same since the first reward is collected after exploration and the plans are optimal thereafter. Note that the number of steps to finish the first task is longer in our case ($18 \pm 0.09$ steps) than EPN, but the average reward in an episode is comparable.

## 4.2 ONE-SHOT STREETLEARN

One-Shot StreetLearn is another task proposed by Ritter et al. (2020) based on the StreetLearn task (Mirowski et al., 2019) to evaluate rapid task solving ability in a more challenging setting with varying connectivity structure across episodes (Fig. 2D). In each episode, the agent is placed in a neighborhood of a city and the task is to navigate to a goal, specified by the goal street view image, and collect the reward. After collecting a reward, the agent is re-spawned in a new location and a new goal is specified. Each episode lasts 200 steps and a new neighborhood is sampled for the next episode. Unlike the MPG, the transition graph changes every episode. We evaluate the ability of CSCG to rapidly learn the model in an episode and to navigate optimally to the goals to maximize the rewards. Note that there is no schema reuse in this setting: we learn a new model for every

episode. This showcases CSCG's ability to learn a model rapidly within a few steps without any prior training and plan efficiently with the learned model.

**CSCG matches optimal planning in One-Shot StreetLearn** For the CSCG agent, we follow an explore and exploit strategy with a hard coded exploration policy. During exploration, agent navigates every action from every observation it encounters while collecting the rewards as it encounters the goals, and uses this experience to learn a CSCG. This is a guided exploration to cover every possible edge in the transition graph. After exploration, agent plans with CSCG and collects rewards (See Appendix B.2 for details). Average reward $\pm$ SEM over 100 episodes is $21.7 \pm 3.7$ (Fig. 2E). This performance is lower than EPN (28.7) as our exploration strategy is not optimal. Since we do not consider optimal exploration in this work, we compare the planning performance on the learned model after exploration. Post exploration, our agent takes on average $4.8 \pm 0.03$ steps, which matches with the optimal value (Ritter et al., 2020) (Fig. 2F). Note that we do not transfer any learning across episodes in this setting since the graph changes every episode, unlike in the MPG. In cities with re-usable StreetLearn graph structures such as grid layout in Manhattan, we can reuse CSCG schemas similar to MPG and benefit from the reuse. We evaluate this schema reuse in detail and in a much harder setting in the following experiments on navigating in rooms with extensively aliased observations.

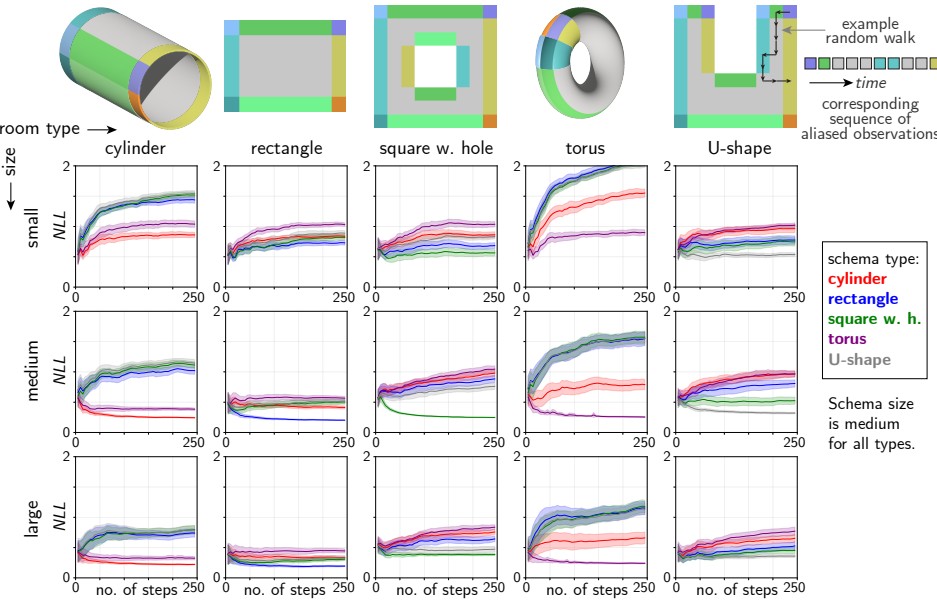

Figure 3: Schema matching in novel environments with size and observation variations. We measure the negative log likelihood (NLL) of observation sequences in a given room under different schemas as the agent does random walks. The schema with the least negative log likelihood (conversely the highest likelihood) is considered the best match. Each panel corresponds to a room of type and size indexed by the column and row headers, respectively. Schemas are trained in medium sized rooms and tested on two other size variations. Error bars are 95% CI of the SEM.

### 4.3 SCHEMA MATCHING IN HIGHLY ALIASED ENVIRONMENTS

In the next set of experiments, we use a more challenging setting with larger environments and extensive aliasing, and evaluate schema matching in novel environments varying in size and observations from the trained environments. We start with a set of environments of different shapes and topologies with extensive aliasing. Analogous to a large empty arena, observations in the interior of these environments are identical (perceptually aliased; Lajoie et al., 2018), see Fig. 3 top row. As described in section 3.2, we learn the schemas in training rooms (Fig. 1D). We evaluate schema matching on test rooms with similar layout but with a novel observation mapping and size variations. In a novel test room, the agent takes a random walk and evaluates the likelihood of the observation sequences given the actions executed under different learned schemas (Fig. 1E). Note that this re-

quires first learning the new emission matrix from the data collected during this random walk and computing the likelihood for each schema. The schema with the best likelihood is considered the matching schema. We evaluate the likelihoods of different schemas at different intervals during the random walks. We used the parameters specified in Appendix B in the following experiments unless otherwise specified.

**CSCG schemas rapidly learn matching bindings in novel environments that differ in size and observations from the training environments.** We generated rooms with different layouts in 3 sizes: cylinder, rectangle, square with hole, torus, and U-shape (Fig. 3). We learn CSCGs for the medium-sized versions of these rooms from a random walk of $50,000$ steps. In the test rooms with novel observation mapping and 3 different sizes per shape, the agent takes a random walk while we learn the new $E$ and evaluate the likelihoods of these observations under all schemas every 5 steps. We compute these values up to 250 steps. We repeat this 25 times in each test room with different random walks. See Appendix B.3 for details. Fig. 3 shows the negative log likelihoods of all test room under all schemas. By reusing clone structure, we are able to correctly match the schema in all cases by 50 steps demonstrating a rapid matching and adaptation to novel environments with size and observation variations (See 8 for results without using clone structure).

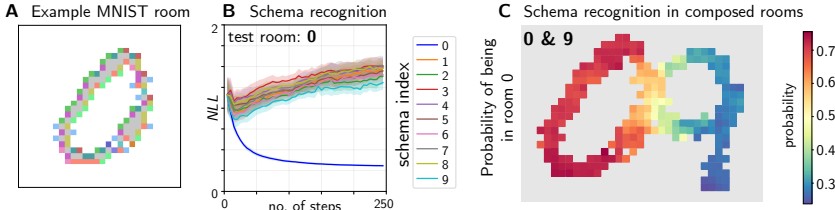

Figure 4: **A.** An example MNIST digit used to train schemas. **B.** Negative log-likelihood (NLL) vs no. of steps in the test room corresponding to digit 0. We are able to recognize the correct schema in fewer than 50 steps. Error bars are 95% CI of the SEM. **C.** The heatmap represents the probability of being in schema 0 at each location in the composed room, averaged over all instances of the agent landing at that location during a random walk of length $20,000$ steps.

**Schema matching in MNIST rooms.** In this experiment, we used a set of ten digits from the binarized MNIST dataset as our room layouts as they provide an interesting variety of shapes and topologies. All the foreground pixels are treated as accessible locations; background pixels are treated as obstacles. Observations at each pixel are sampled in a similar way to the shaped rooms with aliased interiors (see Fig. 4A, Appendix Fig. 9 for examples of the training rooms). We repeat our schema matching experiments as before in this new set of rooms. We first trained schemas on rooms corresponding to all ten digits. To evaluate schema matching, we generated test rooms with the same structure but with new observation mappings. Fig. 4B shows the negative log likelihoods of observation sequences in test room 0 under different schemas (see Appendix Fig. 10 for all rooms). The correct schema was identified in at most 50 steps in all test rooms. (See Appendix B.4 for more details).

## 4.4 SCHEMA MATCHING IN COMPOSED ENVIRONMENTS

To demonstrate how schemas could be used in more complex environments, we constructed test rooms composed of pairs of overlapping MNIST digits (Fig. 4C). As an agent randomly explores a composed room, we compute the likelihood of a sliding window of action-observation pairs under each schema. Note that we use a sliding window here, instead of the entire history of observations, because we are interested in identifying the correct schema at the current time-step. The probability of being in a particular schema can then be computed using the softmax of the log-likelihoods (see Appendix B.4 for more details). In Fig. 4C, we show the probability of being in room 0 at all locations in a room composed of the digits 0 and 9, using a random walk of length $20,000$ steps and sliding window of length 200 steps. We can also compute the accuracy of identifying the correct schema at all locations by thresholding these probabilites. The average accuracy of identifying the correct schema at all locations in all composed test rooms is $88.4 \pm 0.9\%$. See Appendix Table 1 for results on all composed rooms.

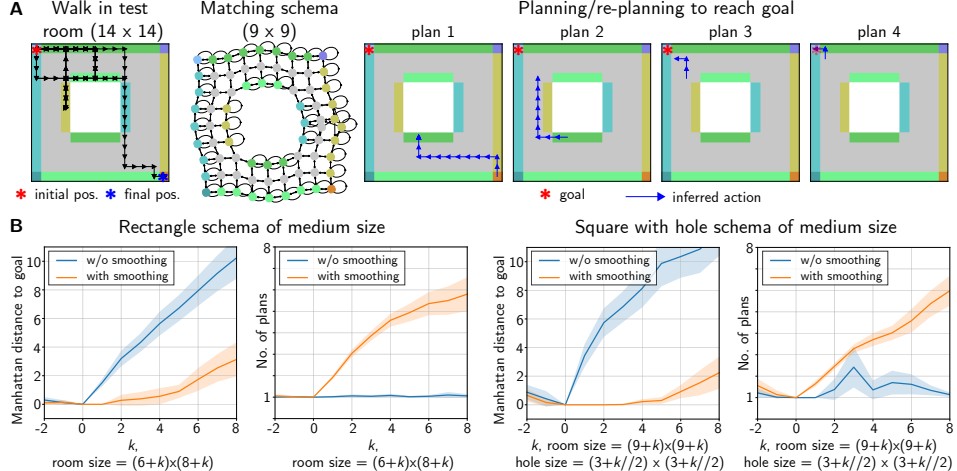

Figure 5: Schemas enable rapid and robust navigation planning in new environments. **A**. Shortcut finding with re-planning in a larger room. Given partial observations in a novel room ($14 \times 14$ square with a $5 \times 5$ hole that is off-center), the best matching schema ($9 \times 9$ square with a $3 \times 3$ hole in the center) can be used to rapidly navigate around obstacles and find the shortest path to a goal. **B**. Shortcut finding performance as measured by the Manhattan distance from the goal and the number of plans/re-plans to reach the goal, as a function of room size. Error bars are 95% CI of the SEM.

## 4.5 RAPID PLANNING IN NOVEL ENVIRONMENTS WITH GRAPH SCHEMAS

Rapid schema matching and binding enables planning in novel environments with limited experience. In this experiment, an agent first executes a random walk in a test room. At the end of the walk, the agent has to find the shortest path to the starting location using the best matching schema. The agent first uses the observations to learn $E$ in the new environment. Next, it uses the schema and the binding $E$ to infer the shortest path to its goal. After executing this plan, if the agent estimates that it has not reached its goal after accounting for the new observations, it is possible that there is a schema mismatch or the estimated $E$ is inaccurate. When this occurs, the agent initiates re-planning. This process is iterated until the agent believes that it has reached its goal after decoding observations from the initial random walk and all subsequent re-planning steps (Fig. 5A).

We evaluate shortcut finding performance in terms of Manhattan distance from the goal location. We used 50 trials with initial random walks of length 200 steps. In Fig. 5B, we show how the performance degrades gracefully as the size mismatch between the schema and the test room increases. We also show how the number of planning attempts required to reach the goal increases as the size mismatch increases. We also evaluated shortcut finding without diagonal smoothing (described in Appendix B). The diagonal smoothing term adds self-transition probability at each node for each action, and allows the agent to account for observation mismatches in aliased settings. Without this smoothing, the agent never reaches the goal in larger size variations of the test rooms.

## 4.6 SCHEMA-BASED NAVIGATION IN SIMULATED 3D ENVIRONMENTS

All our previous experiments focused on 2D grids, but we can use schema-based transfer learning to quickly navigate in novel simulated 3D environments. To demonstrate this, we use a 3D environment (Beattie et al., 2016) with a T-maze layout (Fig. 6A). The agent can navigate with 3 discrete actions (move forward, turn left, turn right) and the observations are RGB images. We use vector-quantization ($k$-means) to get cluster indices from the raw RGB observations and use these indices as input observations to learn a CSCG model of the training environment (Fig: 6B). Fig. 6C shows latent graph learned by the CSCG after training using a random walk. We generate a test 3D environment with the same layout but with different colored walls, floor and environment lighting (Fig. 6D). This corresponds to entirely new RGB observations for an agent navigating in this room. The agent

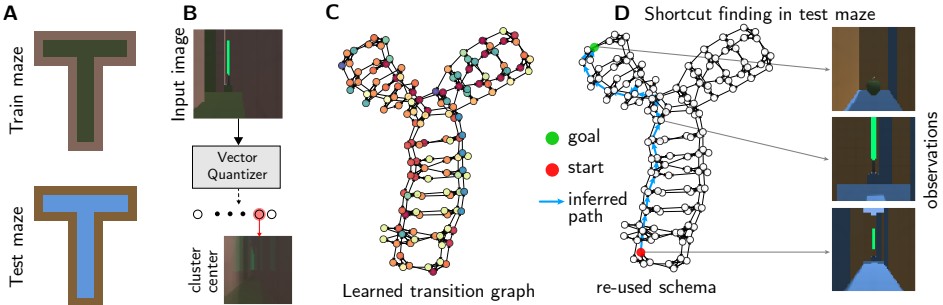

Figure 6: **A.** Top views of the train and test mazes. **B.** Schematic of a vector quantizer used to map input images to a discrete space. **C**. The mapped observations are then used to learn a CSCG for the train maze. **D**. At test time, the agent quickly learns the emission matrix in the new room and uses the schema to navigate to a goal state.

walks in the test room and learns an emission matrix (new bindings for the learned schema) in fewer than 90 steps. Using this model, the agent can plan and navigate to goals in the new room (Fig. 6D).

## 5 DISCUSSION AND FUTURE WORK

Learning abstractions that can rapidly bind to observations from environments that share the same underlying structure is the hypothesized mechanism for transfer learning in humans and animals (Zhou et al., 2021; Tse et al., 2007; Kumaran et al., 2016). We have proposed a concrete computational model for this using graph schemas that learn higher-order structures from aliased observation sequences, and uses a slot-binding mechanism for transferring those schemas to rapidly learn models in new environments.

CSCG schemas learned graph structure in far fewer episodes than a deep RL agent and matched optimal planning in MPG and One-Shot StreetLearn tasks. In highly aliased environments, CSCG schemas found matching schema bindings in novel rooms of different sizes. In composed rooms, we were able to match the correct schemas as the agent moved across rooms of different schemas. We showed successful planning to goals in rooms with shape and size variations from schema by re-planning and updating the model while walking to the goal and this can be adapted to simple 3D environments. There are clear directions for potential future works, some of which are listed below.

**Schema learning from experiences.** We learned schemas independently and explicitly in this work, but in the real world, it might not be feasible to have access to differentiated experiences belonging to distinct schemas. Learning reusable schemas from a continuous stream of experiences could be an interesting future work.

**Schemas vs memories.** We discard previously learned emissions from past experiences and learn a new binding. However, in some cases, previously learned emissions are directly applicable and therefore keeping those in addition could enable even faster zero-shot adaptation when there is a match. This can be thought as keeping specific memories versus using abstract schemas.

**Schema maintenance.** Our schemas in this work are fixed. However, it is possible to update the schemas with new experiences. In fact, children initially tend to perceive and remember experiences that fit in their existing schemas and develop the flexibility later (Piaget & Cook, 1952). Similarly, we could update schemas based on new experiences, and even make the schemas themselves flexible to encapsulate related abstractions but still constrained by rules to allow consistent inference.

**Active exploration.** We used either random or known optimal exploration policies to learn and bind schemas. But the schemas provide action-conditioned beliefs on future observations, and by choosing actions that could optimally disambiguate among different schemas, we could potentially do better than random exploration. Similarly, instead of random exploration to learn a new environment and schema, we could direct the exploration policy by composing known schemas (Sharma et al., 2021), and even actively learn them while exploring.

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

## A    EXPECTATION-MAXIMIZATION LEARNING OF CSCGS

Cloned Hidden Markov Models (HMMs), first introduced in (Dedieu et al., 2019), are a sparse restriction of overcomplete HMMs (Sharan et al., 2017) that can overcome many of the training shortcomings of dynamic Markov coding (Cormack & Horspool, 1987). Similar to HMMs, cloned HMMs assume the observed data $x \equiv \{x_1 \ldots, x_N\}$ are generated from a hidden process $z \equiv \{z_1 \ldots, z_N\}$ that obeys the Markovian property,

$$P(x) = \sum_z P(z_1) \prod_{n=2}^{N} P(z_n|z_{n-1}) \prod_{n=1}^{N} P(x_n|z_n).$$

Here $P(z_1)$ is the initial hidden state distribution, $P(z_n|z_{n-1})$ is the transition probability from $z_{n-1}$ to $z_n$, and $P(x_n|z_n)$ is the probability of emitting $x_n$ from the hidden state $z_n$.

In contrast to HMMs, cloned HMMs assume that each hidden state maps deterministically to a single observation. Further, cloned HMMs allow multiple hidden states to emit the same observation. All the hidden states that emit the same observation are called the *clones* of that observation. With this constraint, the joint distribution of the observed data can be expressed as,

$$P(x) = \sum_{z_1 \in C(x_1)} \cdots \sum_{z_N \in C(x_N)} P(z_1) \prod_{n=2}^{N} P(z_n|z_{n-1})$$

where $C(x_n)$ corresponds to the hidden states (clones) of the emission $x_n$.

Clone-structured cognitive graphs (CSCGs) build on top of cloned HMMs by augmenting the model with the actions of an agent $a \equiv \{a_1 \ldots, a_N\}$ (George et al., 2021),

$$P(x, a) = \sum_{z_1 \in C(x_1)} \cdots \sum_{z_N \in C(x_N)} P(z_1) \prod_{n=2}^{N} P(z_n, a_{n-1}|z_{n-1})$$

### A.1    LEARNING THE TRANSITION MATRIX WITH EMISSIONS FIXED

Learning a CSCG in a new environment requires optimizing the vector of prior probabilities $\pi$: $\pi_k = P(z_1 = k)$ and the action-augmented transition matrix $T$: $T_{ijk} = P(z_n = k|z_{n-1} = j, a_{n-1} = i)$. The emission matrix $E$, which encodes the particular allocation of clones to observations, is kept fixed throughout learning.

The standard algorithm to train HMMs is the expectation-maximization (EM) algorithm (Wu et al., 1983), which in this context is known as the Baum-Welch algorithm. Learning a CSCG using the Baum-Welch algorithm requires a few simple modifications: the sparsity of the emission matrix can be exploited to only use small blocks of the transition matrix both in the Expectation (E) and Maximization (M) steps.

To this end, we assume the hidden states are indexed such that all the clones of the first emission appear first, all the clones of the second emission appear next, etc. Let $N_{\text{obs}}$ and $N_{\text{actions}}$ be the total number of emissions and actions, respectively. The transition matrix $T$ can then be broken down into smaller submatrices $T(u, v, w)$, where $v, w \in \{1, \ldots, N_{\text{obs}}\}$ and $u \in \{1, \ldots, N_{\text{actions}}\}$. The submatrix $T(u, v, w)$ contains the transition probabilities $P(z_n, a_{n-1} = u|z_{n-1})$ for $z_{n-1} \in C(v)$ and $z_n \in C(w)$, where $C(v)$ and $C(w)$ correspond to the clones of emissions $v$ and $w$ respectively.

The standard Baum-Welch equations can then be expressed in a simpler form in the case of a CSCG. The E-step recursively computes the forward and backward probabilities and then updates the posterior probabilities. The M-step updates the transition matrix via row normalization.

**E-Step:**

$$\alpha(1) = \pi(x_1) \qquad \alpha(n)^\top = \alpha(n-1)^\top T(a_{n-1}, x_{n-1}, x_n)$$
$$\beta(N) = 1(x_N) \qquad \beta(n) = T(a_n, x_n, x_{n+1})\beta(n+1)$$

$$\xi_{uvw}(n) = \frac{\alpha(n) \circ T(a_n, v, w) \circ \beta(n+1)^\top}{\alpha(n)^\top T(a_n, v, w) \beta(n+1)}$$

$$\gamma(n) = \frac{\alpha(n) \circ \beta(n)}{\alpha(n)^\top \beta(n)}.$$

**M-Step:**

$$\pi(x_1) = \gamma(1)$$

$$T(u, v, w) = \sum_{n=1}^{N} \xi_{uvw}(n) \oslash \sum_{u=1}^{N_{\text{actions}}} \sum_{w=1}^{N_{\text{obs}}} \sum_{n=1}^{N} \xi_{uvw}(n)$$

where $\circ$ and $\oslash$ denote the element-wise product and division, respectively (with broadcasting where needed). All vectors are $N_{\text{cpe}} \times 1$ column vectors, where $N_{\text{cpe}}$ is the number of clones per emission. We use a constant number of clones per emission for simplicity of description, but the number of clones can be selected independently per emission.

Importantly, CSCGs exploit the sparsity pattern in the emission matrix when performing training updates and inference, and achieve significant computational savings when compared with HMMs (George et al., 2021).

## A.2 LEARNING THE EMISSION MATRIX WITH TRANSITIONS FIXED

With a CSCG, transfer learning between different environments can be accomplished by keeping its transition probabilities $T$ fixed and learning the emissions associated to its nodes $E$ in the new environment. Further, if we know that the new environment preserves the emission structure, then we can further restrict the learning of $E$, with all the rows of $E$ that correspond to the same observation in the original environment sharing the same parameters.

The EM algorithm can be used to learn the emission matrix as follows. The E-step recursively computes the forward and backward probabilities and then updates the posterior probabilities. The M-step updates only the emission matrix.

**E-Step:**

$$\tilde{\alpha}(n) = \left(T(a_{n-1})^\top \alpha_{n-1}\right) \circ E(x_n) \qquad p_\alpha(n) = \sum_{k=1}^{N_{\text{clones}}} \tilde{\alpha}_k(n) \qquad \alpha(n) = \tilde{\alpha}(n)/p_\alpha(n)$$

$$\tilde{\beta}(n) = T(a_n)\left(\beta_{n+1} \circ E(x_{n+1})\right) \qquad p_\beta(n) = \sum_{k=1}^{N} \tilde{\beta}_k(n) \qquad \beta(n) = \tilde{\beta}(n)/p_\beta(n)$$

$$\gamma(n) = \frac{\alpha(n) \circ \beta(n)}{\alpha(n)^\top \beta(n)}$$

**M-Step:**

$$E(j) = \sum_{n=1}^{N} 1_{x_n=j} \gamma(n) \oslash \sum_{n=1}^{N} \gamma(n)$$

Note that the updates here involve $N_{\text{clones}} \times 1$ vectors, where $N_{\text{clones}}$ is the total number of hidden states in the model. $T(a_n)$ corresponds to the transition matrix for the action $a_n$ and is of size $N_{\text{clones}} \times N_{\text{clones}}$, $E(x_n)$ is a column of the emission matrix corresponding to the emission $x_n$, and $1_{x_n=j}$ is an indicator function. The forward message is initialized as $\tilde{\alpha}(1) = \pi \circ E(x_1)$; and the backward message is initialized as a vector of all 1s.

When the clone structure is to be preserved, the $\gamma(n)$ term in the E-step is modified as follows. For each observation $j \in \{1, \ldots, N_{\text{obs}}\}$, we set the posterior probability for all clones of $j$ to be the same:

$$\bar{\gamma}_k(n) = \sum_{l \in C(j)} \gamma_l(n) \quad \forall k \in C(j).$$

For a given $T$ and $E$, the normalization term $p_\alpha(n)$ for the forward messages in the E-step can be used to compute the negative log-likelihood (NLL) of a sequence of observation-action pairs as follows:

$$\text{NLL} = -\frac{1}{N} \sum_{n=1}^{N} \log p_\alpha(n)$$

## B EXPERIMENT DETAILS

In George et al. (2021), it was observed that the convergence of EM for learning the parameters of a CSCG can be improved by using a smoothing parameter called the pseudocount. The pseudocount is a small constant that is added to the accumulated counts statistics matrix ($C$), which ensures that any transition under any action has a non-zero probability. This ensures that the model does not have zero probability for any sequence of observations at test time. During the schema learning phase (learning $T$, with $E$ fixed), we use a pseudocount of $2 \times 10^{-3}$. During the schema matching phase (learning $E$, with $T$ fixed), we use a pseudocount of $10^{-7}$.

In the schema matching phase, we apply an additional smoothing term $\lambda \max(C)$ to the diagonal elements of $C$. This diagonal smoothing term controls the self-transition probability at each node for each action. This is especially important for tasks such as shortcut finding, where we want a schema to adapt to test rooms with size and shape mismatch. For all our experiments that involve learning $E$, with $T$ fixed, we used $\lambda = 0.2$, unless otherwise specified. These values for the pseudocount and the diagonal smoothing were obtained using hyper-parameter sweeps.

### B.1 MEMORY & PLANNING GAME

The game environment is a $4 \times 4$ grid of symbols and the agent can navigate in the four canonical directions by one grid step (up, down, left, right) and collect reward of 1 at a goal location in the grid (Fig. 2A). Reward is 0 otherwise. Once the agent collects the reward at the current goal, the agent is placed in a new random position and a new goal symbol is sampled to which the agent must navigate to collect the next reward. All grid positions have unique symbols and the symbol-position mapping is randomized after each episode, which lasts for 100 actions. The agent's observation is a tuple of symbol in its current position and the goal symbol. See Ritter et al. (2020) for more details. We assume knowledge of collect action function and execute it only when the goal symbol reached.

We employ 3 different hard coded exploration policies to cover the observations in a new episode: random navigation actions, random navigation actions but limited to (up, right), and an optimal set of actions to cover the learned transition graph: a Hamiltonian cycle that repeats (up, up, up, right) four times. For a new episode, we take actions based on one of these policies, learn $E$ at every step, and plan to reach the current goal using the current estimate of $E$. Planning a path from the current position to a goal position is achieved via Viterbi decoding in the CSCG. If the probability of reaching the goal is above a certain threshold, we find the series of actions that lead to the goal. In the beginning, the agent is uncertain about both its own position and the goal position, so it executes actions based on one of the hard-coded policies until we are confident of reaching the goal state from the current state above a probability threshold. If the current goal symbol is not yet observed in this episode, we execute actions based on the policy, otherwise we compute the probability of our planned set of actions reaching the goal symbol. As we navigate, the estimate of $E$ becomes better and the plans are more likely to succeed. Our planning algorithm takes both these uncertainties into account. We evaluated for a total of 100 episodes ($10^4$ steps) and the average reward stays the same after the 9th episode once we learn the schema. Average reward per episode are reported for 100 episodes from 10th episode onwards i.e. after learning.

### B.2 ONE-SHOT STREETLEARN

In each episode of this task, the agent is placed in a new neighborhood of a city and needs to move to a goal location and direct itself to the target direction, specified by the goal street view image, and collect the reward. The agent can take one of four actions: move forward, turn left, turn right, collect reward. After every reward collection, the agent is placed in a new location with a new heading direction and provided a new goal street view image. The agent needs to collect as many

rewards as possible during the episode duration of 200 steps, after which a new neighborhood is sampled for the next episode. StreetLearn environment represents the agent's perception, a street view image, as a unique compressed string that we use as our agent's observation. For the CSCG agent, we follow an explore and exploit strategy. The agent first navigates every action from every observation it encounters while collecting the rewards as it encounters the goals. This is a guided exploration to cover every possible edge in the transition graph. A CSCG model is then learned from this experienced sequence of observations and actions in this episode. For any subsequent step, we find the closest path from the agent's current observation to the goal observation and execute the actions, re-planning at every step to account for any mismatch in model expectation and the reality.

## B.3 SCHEMA RECOGNITION IN ROOMS WITH SIZE VARIATION

We generated rooms of five different types, and three size variations per type, as shown in Fig. 7. We selected the room and barrier dimensions such that the rooms have similar number of accessible states. For the torus and cylinder rooms, we use the same observation map as in the rectangular room case. However, actions wrap around the top-bottom edges for the cylinder, and the top-bottom and left-right pair of edges for the torus, as indicated by the blue arrows in Fig. 7. We learned schemas on the medium size variation of these room types. For each training room, we used action-observation sequences from a random walk of length $50,000$ steps to train the respective CSCG.

For evaluating schema recognition, we used test rooms of all three size variations. Importantly, we generated the test rooms by permuting the observations from the original room. As a consequence, at test time, our model would have to relearn the observation mapping.

In figures 3 (main text) and 8, we show the schema matching performance with and without the use of clone-structure, respectively. We observe that it takes at most 95 steps to identify the correct schema for all test rooms when using clone structure. On the other hand, it takes at most 255 steps to find the best matching schema when the clone structure is ignored. We obtained these results using the paired T-test over 25 random walks, per test room, and a p-value threshold of 5%.

## B.4 MNIST ROOM EXPERIMENTS

We learn schemas for 10 selected MNIST-digits (Fig. 9) using action-observation pairs from random walks of length $100,000$ steps in each room. For experiments with composed MNIST rooms, we selected the sliding window size of 200 by doing a hyper-parameter sweep. The algorithm for schema selection is presented below.

---
**Algorithm 1** Schema selection for composed rooms

---
**Input:** .
  - Schemas $S_1, \ldots S_H$.
  - Observations and actions $x, a$ of length $N$ in the test room $R$.
  - Window size $w$.

  $n \leftarrow w$
  **while** $n <= N$ **do**
      Estimate $E_j$ using $x_{n-w:n}, a_{n-w:n}$ for each schema, $j \in \{1, \ldots, H\}$
      Compute $L_{n,j}$: log likelihood of $x_{n-w:n}, a_{n-w:n}$ under schema $j$
      Compute $p_{n,j} = \text{softmax}(L_n)_j$ : the probability of being in schema $j$ at time step $n$.
      Select schema with highest $p_{n,j}$.
  **end while**

---

In Table 1, we present the accuracy of schema selection for multiple composed rooms.

## B.5 RAPID INFERENCE AND PLANNING USING SCHEMAS

For each room size and shape, we generate a sequence of random actions and repeat every action 3 times. We select a starting location corresponding to a unique observation in the room and execute these actions up to 200 steps and collect the observations. Same action repetition creates trajectories

| Composed digits | Mean accuracy (averaged over five runs) | Standard deviation of accuracy |
|---|---|---|
| Digits 2 and 3 | 87.66 | 4.14 |
| Digits 2 and 7 | 92.47 | 1.51 |
| Digits 0 and 9 | 87.88 | 2.77 |
| Digits 4 and 6 | 85.89 | 2.58 |
| Digits 5 and 1 | 84.28 | 2.74 |
| Digits 5 and 3 | 91.14 | 2.76 |
| Digits 9 and 3 | 85.21 | 3.87 |
| Digits 8 and 7 | 90.86 | 4.04 |
| Digits 6 and 0 | 90.38 | 2.79 |

Table 1: Accuracy of the schema predictions as the agent moves across various locations in composed MNIST rooms. We use the ground truth locations to compute this accuracy. Note that for the composed room experiments, the agent gets two schemas as input ($H = 2$). Further, the accuracy is averaged over five trials of random walks in each room.

where the agent walks farther from the start position with less opportunity to fully explore the surrounding area. The agent is then tasked with going back to the start location of the walk by planning a shortest path back to the start location from the current location. Based on the matched schema, we first learn bindings, an emission matrix $E$, from the observation-action pairs. We then use that model to Viterbi decode the the current state and the start state, and plan our path back to start in the model using max-product message-passing. After executing every action of this plan, we collect the observation and re-plan when the current plan fails to take us the goal state. We repeat this for $50$ such walks for each test room.

## B.6 SIMULATED 3D ENVIRONMENTS

In the simulated 3D environments, observations are $64 \times 64$ pixel RGB images, and the action space is discretized to 3 egocentric actions: go forward by 1m, rotate $90°$ left, rotate $90°$ right. Thus, the agent is only allowed to walk on a grid locations 1m apart. We convert the input RGB images into categorical values using a $k$-means vector quantizer with $k = 36$. For training a CSCG on the example T-maze, we collected a random walk of length $100,000$ steps in the maze.

We also tried other quantization techniques like the vector quantized variational auto encoder (VQ-VAE). We first trained a VQ-VAE, with number of embeddings $= 8$, on images from multiple 3D rooms. An example transition graph of a CSCG trained using the VQ-VAE to generate observations is shown in 11. Even in this case, we observe that we are able to correctly recover the structure of the maze.

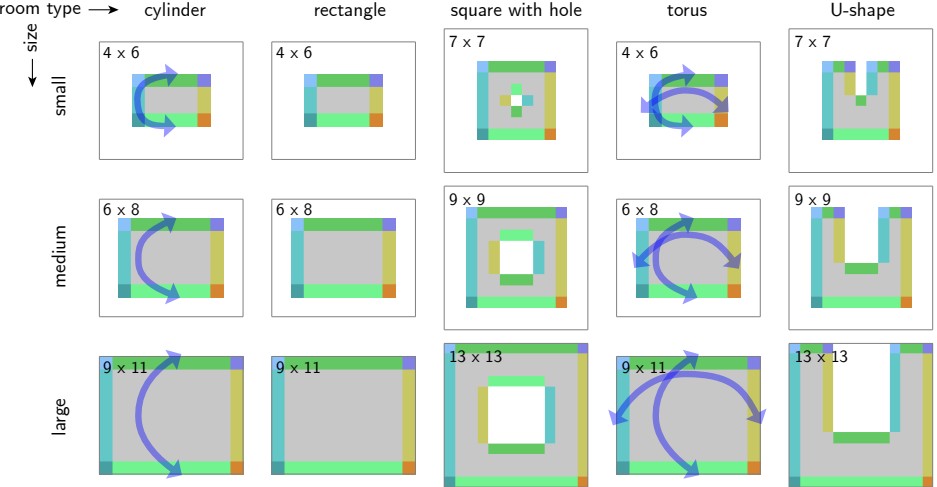

Figure 7: Rooms of different types and sizes used in the schema matching experiments.

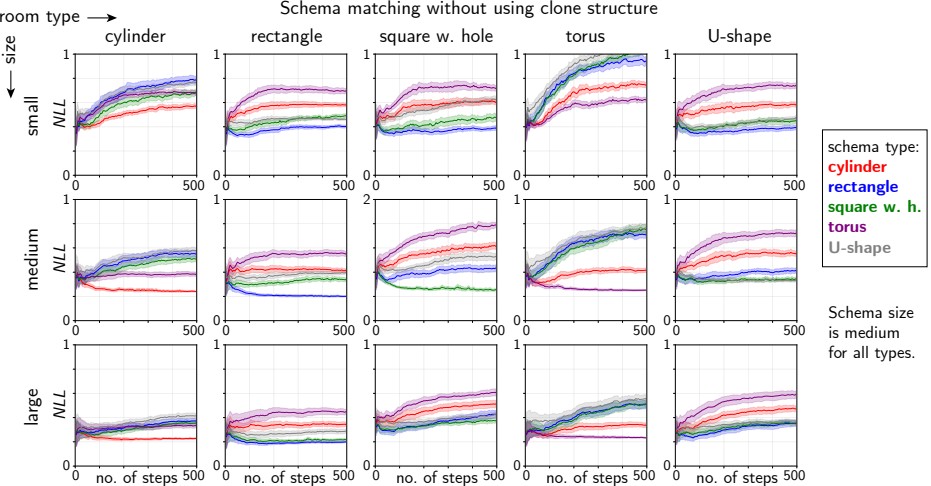

Figure 8: We measure the negative log likelihood (NLL) of observation sequences in a given room under different schemas, without using the clone structure. Each panel corresponds to a room of type, size indexed by the column and row headers, respectively. Error bars are 95% CI of the SEM.

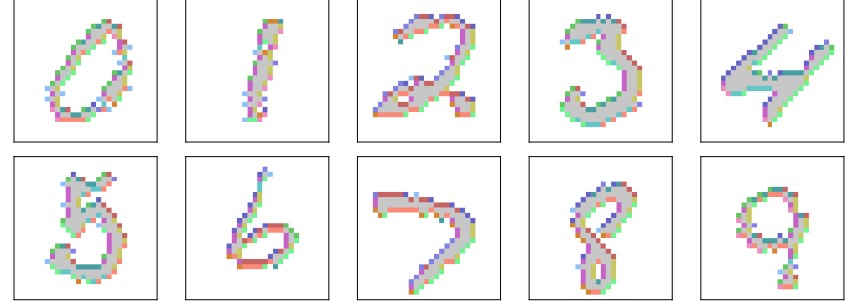

Figure 9: Images from the binarized-MNIST dataset used for our schema matching experiments.

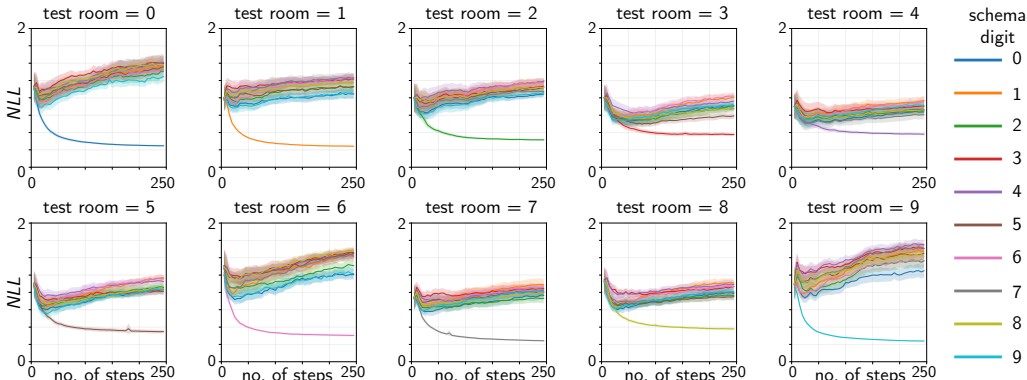

Figure 10: Schema recognition for MNIST rooms. We measure the negative log likelihood (NLL) of observation sequences in a given room under different schemas. Each panel corresponds to a test room indexed by its header. Error bars are 95% CI of the SEM.

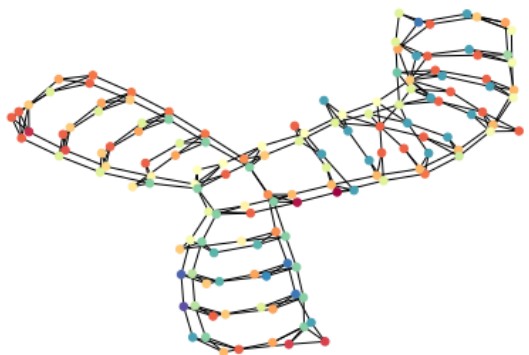

Figure 11: Learned transition graph of a CSCG trained on the T-maze using observations from a VQ-VAE quantizer.

