# OpenReview forum: "Graph schemas as abstractions for transfer learning, inference, and planning"
_ICLR.cc/2023/Conference — Submitted to ICLR 2023_

### Official Review · Reviewer_XbR6 · 2022-10-19

**Confidence:** 3
**Correctness:** 4
**Technical Novelty And Significance:** 2
**Empirical Novelty And Significance:** 2
**Recommendation:** 5

**Clarity, Quality, Novelty And Reproducibility:**

Clarity: Well written, although a bit brief on methodology section.
Quality: Mostly clear.
Novelty: Uses an extensively studied approach (HMMs where we re-estimate part of the model after transfer).
Reproducibility: Should be reproducible.

**Strength And Weaknesses:**

Strength:
* Fast adaptation and transfer in navigation tasks is a relevant and important research topic.
* The paper has extensive experiments.
* The method shows quick adaptation in new settings.

Weaknesses:
* The approach is essentially based on a Hidden Markov Model, where the emission function gets relearned in a new task. This is useful and works well, but Hidden Markov Models and all their variants have of course been extensively studied in previous decades. A major challenge for HMMs is the inability to scale to higher-dimensional problems, and the authors do not really address these concerns (see discussion in conclusion of review).
* It is unclear to me how much prior information these methods need. For example, how is the number of z variables chosen per task? I suppose it usually matches the true number of nodes in the source task, or the number of clusters in the 3D task? This seems to be a strong form of prior knowledge, i.e., the designer already knowns how many nodes to specify.
* After the number of z variables has been specified, they can all possibly be connected in the T matrix right? In other words, the learned topologies that you visualize after learning eliminate all elements of T matrix that get estimated to zero? Or do you also put certain restrictions in the T matrix before training?
* Your methods use hardcoded exploration strategies. You indicate exploration is not part of the current method, which is fine, but it will influence performance. It is unclear to me what exploration strategies the baseline (EPN) uses, and therefore hard to compare performance. In MPG (Fig 2) for example, to what extend is the performance of EPN influenced by a suboptimal exploration strategy? I think these are things you should address.
* In general, you have a very extensive experiments section, which is good, but your methodology section is rather short.

* In StreetLearn (Fig 2), it seems you use pixel based observation in the figure, but I do not read anything about it in the results? Do you use an emission function back to pixel level? I guess you would then need the clustering approach of your last experiment again? Otherwise the task looks much more complicated in the figure than it really is.


**Summary Of The Paper:**

This paper studies ‘clone structured cognitive graphs’ (CSCG), a particular type of action-conditioned Hidden Markov Model (HMM) with specific assumptions on the emission function (deterministic emissions and partial observability, where multiple hidden states can generate the same observation). They investigate the ability of this model to transfer between tasks, primarily when the task topology remains the same but the emission functions change and need to be relearned. Experiments on a range of tasks show that the CSCG model is indeed capable of fast adaptation in transfer experiments.

**Summary Of The Review:**

This paper studies a relevant topic (fast adaptation in navigation tasks), has a lot of experiments, and shows good performance. However, there is one big issue, as mentioned above, which is the use of Hidden Markov Models. Previous literature has extensively shown how HMMs can be specified, estimated, and successfully used in smaller tasks. The authors indeed confirm these ideas, by showing that HMM structure can be efficiently reused between tasks, for example transferring the latent transition structure while re-estimating the emission function.

However, the main issue that HMMs ran into was their inability to scale to more complex, high-dimensional tasks. The authors only touch upon this topic in their final experiment, where they use clustering on pixel observations to make the HMM still applicable in a basic T-Maze. I do not say we need a neural network in every ICLR paper, but I do think this is an issue the paper should address much more. Would it be possible to use the HMM architecture with a learned embedding function? Can you also jointly optimize high-capacity embedding function and latent dynamics, and still transfer one of them afterwards? There is much more work in this direction, see for example [1,2,3,4], and EPN itself of course. I think you need to address these issues, because when it purely comes to the properties of the HMM approach, then I do not think the current paper presents enough novelty.

[1] Marco Fraccaro, Simon Kamronn, Ulrich Paquet, and Ole Winther. A disentangled recognition and nonlinear dynamics model for unsupervised learning. In Advances in Neural Information Processing Systems, pages 3604–3613, 2017.
[2] Matthew Johnson, David K Duvenaud, Alex Wiltschko, Ryan P Adams, and Sandeep R Datta. Composing graphical models with neural networks for structured representations and fast inference. In Advances in neural information processing systems, pages 2946–2954, 2016.
[3] Rahul G Krishnan, Uri Shalit, and David Sontag. Structured inference networks for nonlinear state space models. In AAAI, pages 2101–2109, 2017.
[4] Rangapuram, Syama Sundar, et al. "Deep state space models for time series forecasting." Advances in neural information processing systems 31 (2018).

---

> ### Author Response · Authors · 2022-11-18
> **Response to reviewer XbR6**
>
> We thank the reviewers for their valuable feedback. We addressed some of the concerns raised in common response and addressed more specific comments here.
>
> > It is unclear to me how much prior information these methods need. For example, how is the number of z variables chosen per task?...
>
> Thanks for pointing out this. We do not need to know the exact number of nodes in the ground truth graphs. We only need to assign enough model capacity to recover the graphs. Over allocating the number of states in the model is not a critical issue other than for computational memory reasons and can be better for learning latent representations [1]. In fact, we used 4 times the maximum number of clones needed for any observation in our experiments to learn (i.e > 4 times the number of true states). Even if the learning recovered a graph with extra clones, which likely happens in some cases, it will still be a good generative model of the observations and also work as a good schema to use, in terms of likelihood of observations under the learned model.
>
> > After the number of z variables has been specified, they can all possibly be connected in the T matrix right?...
>
> That is correct. We do not put any more restrictions on the T matrix during learning a schema (CSCG model) once the E is fixed.
>
> > Your methods use hardcoded exploration strategies. You indicate exploration is not part of the current method, which is fine, but it will influence performance. It is unclear to me what exploration strategies the baseline (EPN) uses, and therefore hard to compare performance. In MPG (Fig 2) for example, to what extend is the performance of EPN influenced by a suboptimal exploration strategy? I think these are things you should address.
>
> Baseline method EPN learns a policy for both exploration exploitation combined to solve these tasks. As the reviewer rightly noted, overall reward per episode depends on the exploration policy. However, planning after exploration can be compared between the methods and that is the intention of analysis presented in Fig. 2C and Fig 2F for MPG and Oneshot SteetLearn respectively. These are the same metrics used in the original baseline (EPN) paper to make this same point. For MPG, after training for 10^7 episodes, the model does achieve optimal exploration strategy as indicated by the match with oracle and as stated in the cited work.
>
> > In general, you have a very extensive experiments section, which is good, but your methodology section is rather short.
>
> Thanks for pointing this out. We added detailed descriptions of the model and specific methods in Appendix in the revised manuscript.
>
> > In StreetLearn (Fig 2), it seems you use pixel based observation in the figure, but I do not read anything about it in the results? Do you use an emission function back to pixel level? I guess you would then need the clustering approach of your last experiment again? Otherwise the task looks much more complicated in the figure than it really is.
>
> Following the original work that introduced this task, we used the mapping of each RGB image observation to a hashmap and use that as the observation in our model. So the emissions are these hashmaps that uniquely map to an RGB image of the street view. This is the API provided by the public StreetLearn dataset and also used in the EPN paper public code. The agent’s environment is simplified to walking on the streets by jumping to intersections illustrated in Fig 2D and different views as the agent’s head is aligned to different streets it can traverse. This is the API provided by the one-shot street learn code provided by the EPN paper.
>
> > the main issue that HMMs ran into was their inability to scale to more complex, high-dimensional tasks. ...  I do not say we need a neural network in every ICLR paper, but I do think this is an issue the paper should address much more.
>
> Our motivation in this paper is to demonstrate our method in settings that are simple enough to probe and carefully analyze and reveal the workings of the proposed method, while still showing that they can’t be solved by existing large scale neural network models. To that end, the two baselines we used, Memory & Planning Game and OneShot Street Learn are both simpler than the rest of our experiments (no aliasing and smaller state spaces) and we clearly show that our approach matches the neural networks based methods in performance and requires orders of magnitude less training data (10s vs 10^7 episodes).
>
> We would like to know If the reviewer has any suggestions of specific tasks/benchmarks solved by neural networks approaches to compare our method on those tasks in addition the ones we presented in our paper.
>
>
> **Continued in next comment...**
>
>
> **References**
> 1. Buhai, Rares-Darius, et al. 2020. "Empirical study of benefits of overparameterization in latent variable generative models." ICML.

---

> > ### Author Response · Authors · 2022-11-18
> > **Continued response...**
> >
> > > Would it be possible to use the HMM architecture with a learned embedding function? Can you also jointly optimize high-capacity embedding function and latent dynamics, and still transfer one of them afterwards? There is much more work in this direcgtion, see for example [1,2,3,4], and EPN itself of course. I think you need to address these issues, because when it purely comes to the properties of the HMM approach, then I do not think the current paper presents enough novelty.
> >
> > This is a very good point. We also ran  a version of 3D T-maze  experiment using more powerful VQ-VAEs with a similar result as with much simpler k-means (added to appendix in the revised manuscript). We presented the simpler quantizer in the main paper to show that this doesn’t need a powerful vision model.
> >
> > VQ-VAE can be trained to encode agent’s perceptions into discrete latents on top of which our method works. We trained it independent of the CSCG model using reconstruction loss. Since we are not modeling perception, in a visually complex environment, using a VQ-VAE will be adequate to model the 3D vision and CSCG will work with  the latents supplied by VQ-VAE. Methods such as EPN use a resnet based vision module that is trained end-to-end with the entire network.
> >
> > In our approach, we disentangle perception from sequence learning. By separating the two we can generalize fast learning to varying percepts as long as the task structure remains the same. For example VQVAEs trained in different rendering of 3D mazes could still be used for transfer learning in our approach while it is not clear if an end-to-end trained model can afford such generalization without seeing many such variations during training.
> >
> >
> > We thank the reviewer for extensive related work recommendations and we incorporated these into our related work section as they fit.

---

### Official Review · Reviewer_Zycs · 2022-10-23

**Confidence:** 2
**Correctness:** 4
**Technical Novelty And Significance:** 2
**Empirical Novelty And Significance:** 3
**Recommendation:** 6

**Clarity, Quality, Novelty And Reproducibility:**

The paper is well-written, especially in the experimental part where different experiments are clearly introduced and justified. More details about the model and/or an open implementation would improve quality, clarity and reproducibility.

**Strength And Weaknesses:**

The paper clearly illustrates the strenghts and the limitations of the model on a number of experimental settings designed to highlight different features of the model. I found the presentation of the model in Section 3.1 quite synthetic, and I would have appreciated a more detailed definition of it; also, in Equation (1) I am puzzled by the first \sum whose range is not well specified and by P(z_1).

**Summary Of The Paper:**

The paper proposes and analyzes a computational model for the learning and transfer of schemas as a mean for abstraction. A computational model that decouples learning a schema from observation/emission is suggested, and its performance across a range of empirical simulations is assessed.

**Summary Of The Review:**

The paper provides a good empirical analysis of a computational model using graph schema for abstraction and learning. More discussion about the model and the sharing of the experimental data would enrich this contribution.

---

> ### Author Response · Authors · 2022-11-18
> **Response to reviwer Zycs**
>
> We thank the reviewers for encouraging and constructive feedback.
> Based on this and other reviews, we improved the model description and added more details to the appendix.
> We also clarified the issues pointed out by the reviewers.
>
> CSCG model is already open to public on github and we will put our work on github on the acceptance of the work including the experimental setups.

---

### Official Review · Reviewer_RKUK · 2022-10-26

**Confidence:** 4
**Correctness:** 3
**Technical Novelty And Significance:** 2
**Empirical Novelty And Significance:** 2
**Recommendation:** 3

**Clarity, Quality, Novelty And Reproducibility:**

The paper is clearly written, easy to understand, with limited novelty in terms of the high-level idea but a novel implementation of a graph schema (using CSCGs) and appears to be reproducible.

**Strength And Weaknesses:**

Strengths:

The paper is trying to address a very important research problem. The high-level idea of using graphs as schemas is a very cool concept and I like the overall direction a lot.

Improvements:

1. The work introduces CSCGs as graph schemas for general tasks but only tests it on task with primarily navigation objectives. If the scope of the work is to use graph schemas for only navigation tasks, then this needs to be explicitly stated early in the paper.

2. The related work is very much lacking. References are completely lacking for many related works on using graph-structured schemas to represent long-duration temporal tasks, for much more complex tasks in the real-world in various (supervised, unsupervised and RL) settings. The paper also cites LLMs in first paragraph of section 2 and I'm unsure how they are relevant here. Examples of related works:
A. Multi-modal Cooking Workflow Construction for Food Recipes (https://arxiv.org/abs/2008.09151)
B. Predicting the Structure of Cooking Recipes (https://aclanthology.org/D15-1090.pdf)
C. Procedure Planning in Instructional Videos (https://arxiv.org/abs/1907.01172)
D. Procedure Planning in Instructional Videos via Contextual Modeling and Model-based Policy Learning (https://arxiv.org/abs/2110.01770)
E. proScript: Partially Ordered Scripts Generation via Pre-trained Language Models (https://arxiv.org/abs/2104.08251)
F. Neural Task Graphs: Generalizing to Unseen Tasks from a Single Video Demonstration (https://arxiv.org/abs/1807.03480)

3. The CSCG structure described in section 3 (preliminaries) seems hard to scale. It is basically describing a POMDP setup with nodes as states and edges with transitions corresponding to taking actions. But in most practical problems, we have infinite states so this graph is going to be intractable to maintain, except in small grid worlds.

3. Section 3.1 mentions matrices, so are the state space and observation state considered countably finite? Are the #nodes in graph G known a priori? What about the number of clones per observation?

4. Section 3.1: Do we know the emission model E a priori? If not, how can we keep it fixed throughout learning? Seems like a restrictive and unrealistic assumption to me for most applications!

6. The paper only presents experiments on navigation-style environments in 2D/3D grid-world settings. This implies countably finite states and observations and only 3-4 actions. Hence, the CSCGs are small and tractable. In the case of the 3D T-maze experiment, it is unclear if the vector quantizer used for the 3D T-maze experiments would generalize to more complex 3D environments with complex objects in it.

7. I might have missed the first usage of EPN, but Section 4.1 suddenly brings it up with no prior citation or description. Please clarify if I missed its description in the paper.

**Summary Of The Paper:**

The authors present a method which employs an instantiation of Clone-structured cognitive graphs to learn structured representations of environments, which can be used downstream for transfer, inference and planning. The performance of the proposed approach is evaluated using grid-world style navigation tasks in primarily 2D and in one 3D environment.

**Summary Of The Review:**

Currently the paper has a ton of scope for improvement (see my comments above) and I cannot recommend acceptance. Hence, my overall score is going to be a weak reject.

---

> ### Author Response · Authors · 2022-11-18
> **Response to reviewer RKUK**
>
> We thank the reviewer for very encouraging comments regarding the problem and our approach, and we address specific concerns below in addition to the common response posted.
>
> > The work introduces CSCGs as graph schemas for general tasks but only tests it on task with primarily navigation objectives. If the scope of the work is to use graph schemas for only navigation tasks, then this needs to be explicitly stated early in the paper.
>
> We focused on navigation examples in this work as it provides a clean way to visualize and show the working of the model, and the baselines using neural net models used as similar setup. Elsewhere, CSCGs were shown to effectively model non-navigation sequence learning on varied data such as bird songs, character level language, finite state machines, etc. [1,2]
>
> Using navigation examples in the paper let us test cases of high aliasing with an easy to communicate via visualization the validity of the models, but as the reviewer suspected the approach is applicable to non-navigation tasks.
>
> > The related work is very much lacking.
>
> Given the constraints on space, we focused on closely related work that is relevant for the baselines we used. We will try to add the suggested literature as they fit.
>
> > The CSCG structure described in section 3 (preliminaries) seems hard to scale. It is basically describing a POMDP setup with nodes as states and edges with transitions corresponding to taking actions. But in most practical problems, we have infinite states so this graph is going to be intractable to maintain, except in small grid worlds.
>
> The method is much more scalable than a POMDP due to the particular sparsity pattern of matrix E. For instance, during a forward pass, the complexity per timestep is O(n_clones^2), regardless of the size of the hidden state space, which can be in the billions. While this does not cover infinite states, it can scale to many existing problems (we are beating benchmarks coming from other works). For very large practical applications a more sophisticated, hierarchical representation approach would be needed.
>
> >Section 3.1 mentions matrices, so are the state space and observation state considered countably finite? Are the #nodes in graph G known a priori? What about the number of clones per observation?
>
> Thanks for pointing this out. We do not need to know the number of nodes in the ground truth graphs. We only need to assign enough model capacity to recover the graphs. In fact, we used 4 times the maximum number of clones needed for any observation in our experiments to learn. Even if the learning recovered a graph with extra clones, which likely happens in some cases, it will still be a good generative model of the observations and also work as a good schema to use, in terms of likelihood of observations under the learned model. Recent evidence suggests overparameterization could be beneficial in latent variable generative models [3].
>
> >Do we know the emission model E a priori?...
>
> We only keep the E fixed for learning the schemas. Given a training sequence of observations, we can assign a predetermined number of clones to each unique observation and use that as our emission matrix (all cloned states of an observation emit that observation). We then learn the T matrix with fixed E. During transfer learning to new environment, we keep T fixed and learn new E with or without using the clone structure
>
> > Improvement point 6
>
> Note that our HMM is action-conditioned, so each action results in an independent transition matrix and therefore scales linearly in the number of actions. In the 3D T maze, we have 36 unique observations and ~144 states using simple k-means clustering. In practice, finite state spaces can handle large continuous spaces pretty well: This T maze results in a very small CSCG, so much larger mazes can be managed. At some point while scaling up, a hierarchical approach will be needed. We also ran  a version of this experiment using more powerful VQ-VAEs with a similar result (added to appendix now). We presented the simpler quantizer to show that this doesn’t need a powerful vision model. VQ-VAE can be trained to encode agent’s perceptions into discrete latents on top of which our method works. Since we are not modeling perception, in a visually complex environment, using a VQ-VAE will be adequate to model the 3D vision and CSCG will work with  the latents supplied by VQ-VAE.
>
> Thanks to a careful review, we fixed minor issues noted by them.
>
> **References**
> 1. Dedieu, Antoine, et a. 2019. “Learning Higher-Order Sequential Structure with Cloned HMMs.” arXiv [stat.ML]. arXiv. http://arxiv.org/abs/1905.00507.
> 2. Rikhye, Rajeev V., et al. 2019. “Memorize-Generalize: An Online Algorithm for Learning Higher-Order Sequential Structure with Cloned Hidden Markov Models.” bioRxiv. https://doi.org/10.1101/764456.
> 3.  Buhai, Rares-Darius, et al. 2020. "Empirical study of benefits of overparameterization in latent variable generative models." ICML.

---

### Official Review · Reviewer_rzAg · 2022-10-31

**Confidence:** 3
**Correctness:** 3
**Technical Novelty And Significance:** 2
**Empirical Novelty And Significance:** 1
**Recommendation:** 3

**Clarity, Quality, Novelty And Reproducibility:**

- Quality and novelty
    - The technical contribution of this paper feels, to me, an easy-to-compute subcase of HMMs. In this regard, I didn't find it too novel. If the subcase was very well-justified and how this subcase allows us to be much better/faster than generic HMMs, then I could buy it, but unfortunately I wasn't convinced.
    - The experimental contribution felt a bit weak for two reasons: A) the paper sets to check all the proposed method can do, but little of the different things it cannot do. B) All the domains feel very toy in multiple dimensions (little variability, contrived creation, discrete).
- Clarity
    - Low-level writing was very clear. High-level, I felt too little detail was spent on the proposed model and too much detail on the experiment. The experimental section feels very slow and simple concepts are over-explained. In contrast, I felt I could've received more insights into how the proposed model worked.
    - Small: I would avoid citing the same work in consecutive sentences, it's a bit distracting. This happens a few times in the introduction.
    - In the MNIST experiment: were the domains different digits (i.e. 50k, 80k) or 1 possible image per digit (i.e. 10 different images)?

**Strength And Weaknesses:**

- Strengths
    - I think the paper is clear both in its motivation, what it sets to achieve, and the future remaining directions.
    - Within simulated discrete environments, I believe the types of environments was pretty diverse.
- Weaknesses
    - The overall setting feels quite contrived, the discreteness of the states and actions, the exact clones, etc, are motivated from a cognitive perspective, but the realism of those assumptions isn't well-motivated.
    - Deriving from the previous point, the experiments were quite contrived as well, no realistic experiments were performed, which doesn't allow me to see a way of bringing these ideas to eventual useful systems.
    - The model isn't explained that much and the intellectual novelty compared to standard HMM literature is not explained.

**Summary Of The Paper:**

The paper proposes Graph Schemas, a particular type of action-conditioned HMM with deterministic emissions. The uncertainty comes from the transition matrix, as well as observation "clones", i.e. multiple nodes that output the same observation. These schemas can then be transferred by keeping T fixed while changing E, enabling fast learning and planning in different simulated environments.

**Summary Of The Review:**

The paper reads well and is well-motivated from a CogSci perspective. However, the realism of its final assumptions is not supported, neither by writing or formulation, nor by experiments. To improve the writing I would devote more time on the method section and less on the experiment section. For an ICLR-level bar, I believe there needs to be some non-toy experiment or a very strong motivation. Furthermore, the differences with the HMM literature need to be discussed in much more detail to justify the novelty of this approach.

---

> ### Author Response · Authors · 2022-11-18
> **Response to reviwer rzAg**
>
> We thank the reviewer for their feedback. We respond to individual points below.
> > The overall setting feels quite contrived, the discreteness of the states and actions, the exact clones, etc, are motivated from a cognitive perspective, but the realism of those assumptions isn't well-motivated.
> Deriving from the previous point, the experiments were quite contrived as well, no realistic experiments were performed, which doesn't allow me to see a way of bringing these ideas to eventual useful systems.
>
> - Our motivation in this paper is to demonstrate our method in settings that are simple enough to probe and analyse the workings but still can’t be solved by existing large scale neural network models. To that end, the two baselines we used, Memory & Planning Game and OneShot Street Learn are both simpler than the rest of our experiments (no aliasing and much smaller state spaces) and we clearly show that our approach matches the neural networks based methods in performance and requires orders of magnitude less training data (10s vs 10^7 episodes).
> - As for discrete representations, recent study shows that applying a discretizing bottleneck can improve performance in goal-conditioned RL for solving multiple tasks [1].
> - Control experiments and benchmarks are standard setups designed to test the model characteristics. The two baselines (MPG & OneShot StreetLearn) are proposed by previously published work to specifically test fast task transfer. The rest are designed to be challenging to the existing neural models. Note that the MPG is smaller and simpler than any other setup in our paper.
> - We showed how even the current setup can solve the navigation problem in a 3D continuous world setup by using a vector quantizer for input and discrete actions. We have a version of this experiment using VQ-VAEs which encodes agent’s perceptions into discrete latents on top of which our method works (added to Appendix). This offloads the modeling of 3D vision to any model and models the latents supplied by the model.
>
> > The model isn't explained that much and the intellectual novelty compared to standard HMM literature is not explained.
>
> We addressed this at a high level in our posted common response. We would like to emphasize that the contribution of this paper is a model of abstractions (graph schemas) inspired by cognitive and neurosciences using specific representations that model hippocampus (CSCG) and an approach of using these as schemas for fast knowledge transfer. We added a more detailed description of CSCG in the appendix as they are explained in detail in other works.
>
> >The technical contribution of this paper feels, to me, an easy-to-compute subcase of HMMs. In this regard, I didn't find it too novel. If the subcase was very well-justified and how this subcase allows us to be much better/faster than generic HMMs, then I could buy it, but unfortunately I wasn't convinced.
>
> We addressed this at a high level in our posted common response. Regarding the justification, previous works have shown this model to be applicable to varied domains such as navigation, bird songs, character level language models etc,. all attributed to hippocampus[2-4].
>
> > Clarity:...
>
> We thank the reviewer for the feedback and we cleaned up the issues pointed out.
> For the MNIST, we used 1 exemplar per digit. We used the MNIST digit shapes to evaluate schema matching in individual rooms and rooms composed of multiple schemas.
>
> **References**
> 1. Islam, Riashat, Hongyu Zang, Anirudh Goyal, Alex Lamb, Kenji Kawaguchi, Xin Li, Romain Laroche, Yoshua Bengio, and Remi Tachet Des Combes. 2022. “Discrete Factorial Representations as an Abstraction for Goal Conditioned Reinforcement Learning.” arXiv [cs.LG]. arXiv. http://arxiv.org/abs/2211.00247.
> 2. Dedieu, Antoine, et a. 2019. “Learning Higher-Order Sequential Structure with Cloned HMMs.” arXiv [stat.ML]. arXiv. http://arxiv.org/abs/1905.00507.
> 3. Rikhye, Rajeev V., et al. 2019. “Memorize-Generalize: An Online Algorithm for Learning Higher-Order Sequential Structure with Cloned Hidden Markov Models.” bioRxiv. https://doi.org/10.1101/764456.
> 4. George, Dileep, et al. 2021. “Clone-Structured Graph Representations Enable Flexible Learning and Vicarious Evaluation of Cognitive Maps.” Nature Communications 12 (1): 2392.

---

### Author Response · Authors · 2022-11-17
**Response to common concerns**

We thank the reviewers for the feedback. We would like to address some common issues raised and clarify some misconceptions. Overall, we believe the novelty of our approach is in proposing a model of abstractions inspired by cognitive and neurosciences using specific representations that model hippocampal representations (CSCG) and an approach of using these as schemas for fast knowledge transfer. We address specific points below:

**Novelty of CSCG and graph schemas**
1.  It is known that constraining HMMs in particular ways can lead to faster and more effective learning due to the avoidance of local minima \[1\]. CSCG is novel in having discovered a constraint that makes overcomplete HMMs learn much faster. By having multiple hidden states tied to the same emission, it bypasses the learning of the emission matrix and more effectively learns the transition matrix. Combining this with the observation that overparameterization of latent states helps \[2\], makes for effective learning in CSCG. Of course, the idea looks simple in hindsight, but dismissing this novelty would be equivalent to dismissing the novelty of weight tying in convolutional neural nets. The weight-tying in CSCG is analogous to a convolution in the latent space.
2.  Treating the learned transition matrix as fixed, and then learning the emission matrix, uses the same idea of constrained HMMs \[1\], but in a complementary way to keeping the E matrix fixed and learning the transition matrix.
3.  Slot structures in graphs: The reviewers comment as if ‘graph schemas’ is an established idea. In our knowledge, this is the first instantiation of the idea on how to have slots in the graph, and how to match them to observations in a way that enables transfer of knowledge.
4.  Taken together these ideas are novel, and non-trivial to arrive at. First, using a fixed Emission matrix to learn the graphs, and then using the graphs as fixed to learn the emission matrices uses the insights from the Roweis (1999) paper in two different ways.
5.  Transferring graph representations is considered an important problem in abstraction \[3\]. People who have tried to solve similar problems earlier were discouraged by trying to solve a graph correspondence problem \[4-6\] whereas our formulation side steps this difficulty by finding approximate graph isomorphisms using probabilistic inference.

**Properties of over-parameterized over complete HMMs**
1.  The model relies on beneficial properties of overparameterized and overcomplete HMMs. This is an area that is of theoretical and practical importance \[7\].

**Representational power of HMM for learning discrete graphs.**
1.  The problem we tackled was that of learning and transferring discrete graphs from aliased sensory observation. Alternating constrained HMM structures are used to tackle the learning, and graph matching parts of this problem. All graphs used in the real world have a finite number of states. Therefore finite graph learning itself is an important and interesting problem. We disagree with the characterization that only infinite states are interesting.
2.  We appreciate the statements about HMMs and their lack of distributed representations, but those are not needed in the problems we tackle. We argue that judgements about the suitability of a model should be made based on the problem context. For example, in the context of navigation, the latent states are locations. A discretization of the locations is an approximation. Within that approximation, the HMM formulation of graph learning has no inherent combinatorial explosion.
3.  The overall model is a composition of graphs, whose emission matrices are determined on the fly. While it is possible to reduce every model to a flat HMM (Factorial HMMs and Hierarchical HMMs are just HMMs), it ignores the representational and learning complexity. The specific instantiations of how to constrain the graph learning, and how to quickly match graphs in new settings is important and cannot be obtained as a trivial extension of an HMM.

**References**
1.  Roweis, Sam. 1999. “Constrained Hidden Markov Models.” Advances in Neural Information Processing Systems.
2.  Buhai, Rares-Darius, et al. 2020. "Empirical study of benefits of overparameterization in latent variable generative models." International Conference on Machine Learning.
3.  Shanahan, Murray, & Melanie Mitchell. 2022. “Abstraction for Deep Reinforcement Learning.” http://arxiv.org/abs/2202.05839
4.  Creswell, Antonia, et al. 2021. "Unsupervised object-based transition models for 3d partially observable environments." Advances in Neural Information Processing Systems
5.  Greff, Klaus, et al "On the binding problem in artificial neural networks." arXiv preprint arXiv:2012.05208 (2020).
6.  Crouse, Maxwell, et al. 2021. “Neural Analogical Matching.” AAAI.
7.  Sharan, Vatsal, et al. 2017. "Learning overcomplete hmms." Advances in Neural Information Processing Systems

---

### Decision · Program_Chairs · 2023-01-20

**Decision:**

Reject

**Justification For Why Not Higher Score:**

The authors need to show that the algorithms also scales to higher dimensionality.

**Justification For Why Not Lower Score:**

N/A

**Metareview: Summary, Strengths And Weaknesses:**

The paper proposes graph schemas which are a simplified version of an HMM. Here, the authors learn the transition matrix and keep it fixed for several tasks while the emmision models are adapted for each task. The model has been shown to exhibit fast learning and planning in different simulated environments.

The paper is well written and looks at a diverse set of environments and applications. However, the setting and the experiments feel quite contrived for some reviewers resulting in a poor evaluation. Reviewers were also concerned about realism of the used assumptions as well as too simple experiments that did not go beyond the level of toy tasks. Also, the method section in the paper needs to be extended. The authors provided an extensive response. Unfortunately, none of the reviewers engaged in the discussion leaving the scores unchanged. While the authors address many of these concerns adequately, I think it remains to be shown that the method scales also to higher-D cases. As no reviewer was very enthusiastic of the paper, I recommend rejection.

**Summary Of Ac-Reviewer Meeting:**

N/A